# The Uremic Toxin Indoxyl Sulfate Accelerates Senescence in Kidney Proximal Tubule Cells

**DOI:** 10.3390/toxins15040242

**Published:** 2023-03-25

**Authors:** Yi Yang, Milos Mihajlovic, Manoe J. Janssen, Rosalinde Masereeuw

**Affiliations:** Division of Pharmacology, Utrecht Institute for Pharmaceutical Sciences, Utrecht University, 3584 CG Utrecht, The Netherlands

**Keywords:** kidney fibrosis, conditionally immortalized proximal tubule epithelial cells, senescence-associated secretory phenotype (SASP), transcriptome

## Abstract

Kidney fibrosis is the common final pathway of nearly all chronic and progressive nephropathies. One cause may be the accumulation of senescent cells that secrete factors (senescence associated secretory phenotype, SASP) promoting fibrosis and inflammation. It has been suggested that uremic toxins, such as indoxyl sulfate (IS), play a role in this. Here, we investigated whether IS accelerates senescence in conditionally immortalized proximal tubule epithelial cells overexpressing the organic anion transporter 1 (ciPTEC-OAT1), thereby promoting kidney fibrosis. Cell viability results suggested that the tolerance of ciPTEC-OAT1 against IS increased in a time-dependent manner at the same dose of IS. This was accompanied by SA-β-gal staining, confirming the accumulation of senescent cells, as well as an upregulation of p21 and downregulation of laminB1 at different time points, accompanied by an upregulation in the SASP factors IL-1β, IL-6 and IL-8. RNA-sequencing and transcriptome analysis revealed that IS accelerates senescence, and that cell cycle appears to be the most relevant factor during the process. IS accelerates senescence via TNF-α and NF-ĸB signalling early on, and the epithelial-mesenchymal transition process at later time points. In conclusion, our results suggest that IS accelerates cellular senescence in proximal tubule epithelial cells.

## 1. Introduction

The accumulation of senescent cells is a feature of chronic kidney disease (CKD) and contributes to the progression of the endpoint kidney fibrosis [1]. Acute (short-term) senescent cells can be cleared by the immune system, through chemo-attracting of immune cells, followed by tissue regeneration, while chronic (long-term) senescent cells establish a pro-inflammatory environment and aggravate the disease [2,3]. Senescent cells can be identified by their permanent cell cycle arrest, proliferation limitation and secretion of senescence-associated secretory phenotype (SASP) factors [4].

Cell cycle arrest in senescence is mainly executed by two pathways: the p53/p21^CIP1/WAF1^ (p21) and p16^Ink4a^ (p16)/pRb checkpoint pathways [5], which are activated independently during the senescence process. First, p53 is activated following DNA damage, promoting a p21-dependent cell-cycle arrest [6,7]. Second, p16 prevents the action of the cyclin dependent kinases by suppressing Retinoblastoma 1 (pRb), thus leading to cell cycle arrest [8].

Chronic (long-term) senescent cells are alive, metabolically active and show resistance to apoptosis, which is a common feature of proliferation limitation. Two main pathways contribute to apoptosis-resistance. One is the intrinsic pathway resulting from mitochondrial dysfunction and involving Bcl-2 family members [9]. The abnormal regulation of anti-apoptotic and pro-apoptotic Bcl-2 family proteins triggers the senescent cells to be in a primed apoptotic state, thus making cells stay alive without undergoing proliferation or apoptosis [10]. The other is the extrinsic pathway, regulated by death receptors [11]. Death receptors interact with death ligands, activating caspase proteins and inducing apoptosis [9]. Death receptors, such as tumour necrosis factor receptors (TNFR) (e.g., TNFR1 and TNFR2) and their ligands (e.g., TNF-α), are also found to be important SASP factors [12,13]. Senescent cells primed by TNF-α amplify the senescence process via a paracrine effect [14]. The alteration of intrinsic and extrinsic pathways finally results in apoptosis resistance and senescence [15,16].

SASP factors include proinflammatory and/or profibrotic compounds, such as numerous cytokines (e.g., IL-1β and IL-6), chemokines (e.g., CXCL8 and CCL2), growth factors (e.g., CTGF and TGF-β) and matrix-metalloproteinases (MMPs) [17,18]. The factors are involved in various pathways to maintain and reinforce senescence, but are also key players in its transmission and can undergo paracrine signalling [19,20].

Uremic toxins are endogenous metabolites that accumulate systemically during kidney function decline. In healthy conditions, the waste solutes are renally cleared through glomerular filtration or active secretion by the proximal tubular epithelial cells [21]. Protein-bound uremic toxins, such as indoxyl sulfate (IS), are highly retained during kidney failure, poorly removed by dialysis therapy, and associated with comorbidities in CKD [22]. IS is taken up by proximal tubular cells via organic anion transporter 1 and 3 (OAT1 and OAT3) [23,24]. It was reported that uremic toxins (including IS) can induce proximal tubular injury via OAT1-mediated uptake [24], and can regulate the secretion of organic anions through AhR and EGFR mediated signaling [25]. Furthermore, IS displays proinflammatory and prooxidative effects during CKD, stimulating the expression of SASP factors [26,27]. This uremic toxin may contribute to senescence through the generation of reactive oxygen species (ROS), and p53 and NF-ĸB signalling [28,29,30]. It has been demonstrated that IS also upregulates the SASP factors MCP-1 and ICAM-1 in kidney epithelial cells via p53 and NF-ĸB signalling [31,32]. We previously showed that IS activates the inflammasome NLRP3, thereby inducing the expression of IL-1β with an increase in ROS [33]. Here, we hypothesize that IS maintains and accelerates senescence by regulating SASP factors.

We recently demonstrated that a human conditionally immortalized proximal tubule epithelial cell line overexpressing the organic anion transporter 1 (ciPTEC-OAT1) acquires a conditional senescent phenotype when cells are cultured at a non-permissive temperature [34]. Considering the facts that the main focus of the present work is the effect of IS in kidney senescence and that (primary) PTEC rapidly lose the expression of OAT1 when cultured in vitro, the ciPTEC-OAT1 cell model, showing stable OAT1 expression and function, was used in this study [35]. In this experimental in vitro model, p53 and p21 are active at a permissive temperature (33 °C) through the expression of the temperature-sensitive mutant U19tsA58 of the SV40 large T antigen (SV40T). Furthermore, the cells show a senescence-like arrest when switched to a non-permissive temperature (37 °C) [36].

In this study, we employed ciPTEC-OAT1 as a model to investigate whether IS accelerates the senescence phenotype. Further, transcriptome analysis was conducted to understand the underlying mechanisms.

## 2. Results

### 2.1. Proximal Tubule Cells Show Increased Cellular Tolerance against Indoxyl Sulfate over Time

To understand the appropriate time point for senescence acceleration of IS, we used clinically relevant IS concentrations for up to 9 days and confirmed cellular senescence via SA-β-gal staining. We previously demonstrated that IS exerts an IC_50_ value for cell viability of 2.0 mM after 24 h exposure, which is beyond clinically relevant IS concentrations (on average in severe uremia approx. 200 µM [37]). Dox was employed as a control due to its known senescence inducing effect and regulation of cell death pathways [38,39,40]. The tolerance of ciPTEC-OAT1 to IS increased over time when maintaining the IS concentrations (Figure 1A). 

In fact, cell viability at Day 6 (for 200 µM IS) and Day 9 (for 200 and 400 µM IS) was significantly different when compared to Day 3, while Dox (Figure 1B) did not show differences amongst the three time points. The half maximal inhibitory concentrations (IC_50_, described in the legend), as a measure of the potency of IS and Dox, showed a dose- and time-dependent cell viability reduction. SA-β-gal staining confirmed the accumulation of senescent cells (Figure 1C,D). The results suggest that doxorubicin not only induces senescence but also eliminates senescent cells, while IS-induced senescence is more stable.

### 2.2. Indoxyl Sulfate Exposure Results in Sustained Expression of Common Senescence Markers

To investigate further the senescence phenotype development after IS treatment, we determined the gene and protein expression levels of p21 and LaminB1 as known senescence hallmarks (Figure 2). Compared to the non-senescent group (Day 0), LaminB1 was downregulated at 37 °C at Day 9 for protein level (Figure 2A,B) and on Day 6 and Day 9 for mRNA level (Figure 2F), whereas p21 was upregulated at 37 °C for both protein level (Figure 2C,D) and mRNA level (Figure 2E) at different time points. These findings suggest that the cells obtained a senescence phenotype over time, consistent with our previous results [34]. This effect was sustained when cells were treated with IS, as shown by clear differences and upregulation and downregulation trends (Figure 2A–F).

### 2.3. Indoxyl Sulfate Accelerates Common SASP Factors Secretion

SASP factors maintain and reinforce senescence. Here, we evaluated the secretion of typical SASP factors (IL-6, IL-8 and IL-1β) to further investigate the effect of IS on senescence development. All SASP factors were upregulated over time at 37 °C in ciPTEC-OAT1 cells during maturation (Figure 3). Following exposure to IS, IL-6 was upregulated on Day 3 at all concentrations (Figure 3A).

Similarly, as shown in Figure 3B, the levels of IL-8 were elevated by IS on Day 3 and further increased to equally high levels of IL-6 and IL-8 at Day 6 and Day 9, a process that was also observed for IL-1β (Figure 3C). Our results suggest that IS accelerates high secreted levels of common SASP factors during senescence development.

### 2.4. Transcriptomic Analysis Reveals Large Scale Changes in Gene Expression during Maturation

Figure 4 shows the transcriptome analysis and an overview of the responses of ciPTEC-OAT1 upon culturing in the absence or presence of IS. PCA plots demonstrate that the main source of variation across the transcriptome is linked to the maturation time (Figure 4A), and PC1 is mostly composed of the genes involved in cell cycle-related processes (Appendix A). Further, there are significant correlations between protein levels and the mRNA abundance of IL-6, IL-8 and IL-1β, validating the transcriptome analysis (Appendix A). Based on the expression patterns, several gene clusters can be identified, and the genes for whose expression goes down over time (Cluster A) and can be linked to pathways associated with cell growth and cell division. On the other hand, genes that go up over time (Clusters E and F) contain genes linked to inflammation and epithelial to mesenchymal transition (Figure 4B–D). This is in line with what we expect for cells undergoing senescence and cell cycle arrest, with a clear change between Day 1 and Day 3. A list of enriched pathways (from KEGG and MSigDB Hallmark analyses) and associated differentially expressed genes can be found in Appendix A.

### 2.5. IS Accelerates Senescence by Regulating Senescence Markers

During maturation for 9 days, many senescence related genes are changed (Figure 5A). The upregulation of SASP factors and p21 observed is consistent with our previous findings [34]. The expression levels of common senescence markers are shown in Figure 5B. Compared with the non-senescence group (Day 0), p21 (*CDKN1A*) was upregulated from Day 1 onwards; β-gal (*GLB1*) was upregulated on Day 3 and Day 9, whereas a downregulation of LaminB1 (*LMNB1*) was observed at day 1 in the IS-treatment group and in all groups at later time points.

The upregulation of common SASP factors is shown in Figure 5C, with upregulations observed for IL-8 (*CXCL8*) at Day 3 in the IS treatment group, IL-6 (*IL6*) in both the IS-treated and non-treated group at Day 3, IL-1β (*IL-1B*) and MMP-1 (*MMP1*) in IS-exposed groups at Day 3 and Day 9, IL-1α (*IL-1A*) in IS treatment groups at Day 1 and Day 9, and MMP-3 (*MMP3*) at Day 9 after IS exposure. SASP factors such as IL-8, IL-1β, IL-1α and MMP1 showed significant upregulations on Day 3 and Day 9, IL-6 showed remarkable upregulation on Day 3, and MMP3 had significant upregulation on Day 9.The observed trend in IS-mediated upregulation of p21, decrease of LaminB1 and increase of IL8, IL-1α, IL-1β, MMP-1 and MMP-3 at Day 9 would suggest that IS accelerates senescence by regulating common senescence markers, SASP factors and epithelial-mesenchymal transition (EMT) markers.

### 2.6. IS Induces Expression of SASP Factors and EMT Markers

The enrichment pathways from the MSigDB Hallmark of DEGs indicate that IS accelerates senescence by regulating the genes involved in TNF-alpha Signalling via the NF-ĸB (TNF-α/NF-ĸB) pathway at early time points (Days 1 and 3), and genes involved in the EMT process between Day 3 and Day 9 (Figure 6). A list of differentially expressed genes in these pathways in the presence of IS can be found in Appendix A.

## 3. Discussion

Cellular senescence is identified as cell cycle arrest and the limitation of cell proliferation, which is related to kidney disease and fibrosis [41]. Chronic (long-term) senescent cells accumulate and finally aggravate the disease [2,3]. Senescent cells excrete SASP factors, which are key players in the paracrine induction of secondary senescence or senescence transmission [19]. IS, as a protein-bound uremic toxin, is a well-known proinflammatory and prooxidative metabolite with negative effects in CKD [26,27]. IS is poorly removed by dialysis therapy, and associated with many comorbidities in chronic kidney dysfunction [22]. We previously proved that ciPTEC-OAT1 maturation can be employed as a senescence model and described how IS induces inflammasome-mediated IL-1β production [33,34]. In the present study, we demonstrated that IS accelerates senescence in ciPTEC-OAT1 as seen by senescence phenotypes.

Our cell viability results demonstrated that cells become tolerant to IS exposure over time in a concentration-dependent manner, accompanied by increased SA-β-gal expression and activity. This suggest that this uremic toxin accelerates senescent cell accumulation at relevant uremic concentrations, as the average IS plasma concentration in CKD patients is around 200 µM [33]. In contrast, Dox reduced cell viability dose- and time-dependently. Dox is a chemotherapeutic agent that regulates cell death pathways [40], and is often used to induce senescence in experimental research [38,39]. Our findings are in agreement with Dox-eliminating senescent cells, as reflected by the decreased number of senescent cells with increasing dose and exposure time.

Furthermore, the upregulation trend of p21 [42] and downregulation trend of LaminB1 [43] is consistent with the induction of senescence. p21 is a cyclin-dependent kinase inhibitor, and its upregulation is indicative of cell cycle arrest at either G1/S or G2/M checkpoints [44]. The p53/p21 pathway plays a key role in the initiation of senescence [45]. We previously demonstrated that senescence is mainly induced via the p53/p21 pathway in ciPTEC-OAT1 [34]. Here, our findings suggest that IS accelerates or maintains the expression of p21 at a high level [46], which results in a permanent cell cycle arrest and leads to chronic senescence. LaminB1 regulates cell proliferation and senescence via the mitochondrial ROS signalling pathway [43]. Loss of LaminB1 induces and accelerates senescence [43,47], and its decreased expression was observed earlier in ciPTEC-OAT1 upon culturing [34]. Current results suggest that there is a decreasing trend of LaminB1 levels after IS treatment. There are some discrepancies between mRNA and protein levels of the results. This is because the changes in mRNA levels do not always reflect protein levels directly, due to translation, but may influence other protein modification processes and/or various levels of regulation between the transcript and the protein product.

IL-6, IL-8 and IL-1β are typical proinflammatory mediators released in senescence, and are referred to as SASP factors [17,18,48]. IL-6 and IL-8 induce senescence and inflammation, which promote paracrine senescence [49]. IL-1β produced by senescent cells follows IS-induced and ROS-mediated activation of NLRP3 [33,50], thus promoting inflammatory stress and reinforcing senescence [51]. IS is known to stimulate the expression of IL-6 and IL-1β both in vitro and in vivo [52,53] and upregulation of IL-8 in vitro [54]. Current results suggest that IS induces IL-1β expression time-dependently and promotes IL-6 and IL-8 expression early on during the process. SASP factors are involved in various pathways to maintain and reinforce senescence, and are key players in senescence transmission through paracrine signalling [19,20]. The expression level of SASP factors is higher or kept at a high level in an IS treatment group compared to a no-treatment group; therefore, we believe that IS maintains and reinforces senescence.

To obtain an overview of the most important molecular changes underlying the role of IS in acceleration of senescence, bulk-transcriptomic analysis in ciPTEC-OAT1 cells exposed to IS for up to 9 days was performed. The results suggest that the main source of variation across the transcriptome is the maturation time. Gene expression linked to cell cycle (KEGG pathways) and G2/M checkpoints (MSigDB Hallmark) go down, confirming cell cycle arrest. Senescence has a permanent cell cycle arrest in the G1 or possibly the G2 phase of the cell cycle [4,55]. High expression of p21 is known to be responsible for G2/M arrest in human renal proximal tubular cells, accompanied by the loss of laminB1 [56,57], which is in line with results of this study.

We demonstrated previously that ciPTEC-OAT1 becomes senescent after 9 days maturation at a non-permissive temperature (37 °C) [34], and the transcriptomic results aid in exploring the underlying mechanisms further. Different pathways are involved in senescence process, including FOXO, mTOR, p53 and calcium signalling pathways, which results in cell cycle arrest and accumulation of SASP factors. The expression of senescence markers (p21, β-gal and LaminB1) and SASP factors (IL-6, IL-8 and IL-β) evaluated by transcriptomic analysis is consistent with our findings described earlier. IL-1α, MMP-1and MMP-3 are also important SASP factors [18]. IL-1α regulates other SASP factors, such as IL-6 and IL-8, promoting senescence [58]. MMP-1and MMP-3 belong to the MMP family. MMPs shed ectodomains of cell surface receptors and activate other SASP factors [59], thus regulating the extracellular matrix and promoting EMT and kidney fibrosis [60].

Transcriptome analysis shows that IS exposure increases expression of SASP factors and EMT markers in the senescence process. Those genes are mainly involved in TNF-α/NF-ĸB pathway at early time points (Day 1), and in the process of EMT at Day 9. The TNF-α/NF-ĸB pathway is a crucial mediator of inflammatory and immune responses [61], while EMT is an important cellular programme that regulates embryogenesis and wound healing processes, and it is commonly active after kidney injury [62,63]. Both pathways are known to promote senescence and are involved in modulation, production and/or release of proinflammatory and profibrotic SASP factors [4,64,65,66]. IS has been described to contribute to both inflammation and fibrotic processes in CKD via SASP factors [18,67,68,69], which is in line with the current results.

In conclusion, the present results indicate that IS may contribute to kidney disease by accelerating senescence, through the regulation of senescence markers and by modulating inflammatory and profibrotic processes, as evidenced by changes in the TNF-α/NF-ĸB pathway and the EMT process (Figure 7).

Our study has its limitations as we only used one model of senescence in vitro. However, there is currently no other cell model available that has features that come close to adult human PTEC. An animal study would not yield more insight into the uremic toxin-induced effects either, as the kidney pathology in animals does not fully reflect human kidney etiology and is limited by ethical considerations. Thus, a perfect model does not exist, and one has to compromise on the inclusion of some details of kidney senescence. Our current study focused on the contribution of IS in senescence to investigate the possible pathways that IS may be involved in, and paving the way for future experiments using 3D and in vivo models of chronic kidney disease to test therapeutic interventions such as senolytics.

## 4. Materials and Methods

### 4.1. Culture ciPTEC-OAT1 

The ciPTEC-OAT1 [35] were cultured in a complete culture medium, consisting of phenol-red free DMEM-HAM’s F12 medium supplied by Gibco (Paisly, UK), supplemented with 5 μg/mL insulin, transferrin and selenium separately, 35 ng/mL hydrocortisone, 10 ng/mL epidermal growth factor, 40 pg/mL triiodothyronine and 10% (*v*/*v*) fetal calf serum (FCS) purchased from Greiner Bio-One (Alphenaan den Rijn, The Netherlands). Cells were cultured until 90% confluence at 33 °C, and then transferred to 37 °C with 5% (*v*/*v*) CO_2_ to mature for up to 9 days coculturing with or without IS and the medium was replaced every three days. Cells were seeded in 96-well format plates for cell viability assay, and in 6-well plates for Western blot, Elisa, Real-time PCR and RNA-Seq.

### 4.2. Cell Viability Assay

PrestoBlue^®^ cell viability reagent was used to test cell metabolic activity, which was purchased from Thermo Scientific (Vienna, Austria). The ciPTEC-OAT1 were cultured at 33 °C, and then matured at 37 °C for up to 9 days. Expose cells to increasing concentrations of indoxyl sulfate (IS) and doxorubicin (Dox) during maturation. Cell viability was evaluated at 0, 3, 6 or 9 days. IS was purchased from Sigma-Aldrich (Zwijndrecht, The Netherlands), Dox was obtained from MedchemExpress (Huissen, The Netherlands). GraphPad Prism (version 9.3.0, La Jolla, CA, USA) was used to obtain the cell viability curve.

IC_50_ values (inhibitory constants at 50% of control viability levels) were used to express cell viability. The results of cell viability were plotted via log IS and Dox concentration versus-viability with background subtraction. The normalized data were fitted using nonlinear regression with a variable slope restricting the bottom to 0. Six replicates of a minimum of three different experiments were performed.

### 4.3. SA-β-Gal Staining Assay

After the exposure, cells were washed twice with Hanks’ Balanced Salt Solution (HBSS; Gibco, Life Technologies, Paisly, UK). An SA-β-gal staining assay was performed according to the manufacturer’s instructions. A Senescence Detection Kit was used to identify the SA-gal-positive cells (ab65351, Abcam, Bristol, UK). The staining was evaluated for the appearance of a blue colour through an optical microscope (200× total magnification).

### 4.4. Western Blot

The ciPTEC-OAT1 were exposed to increasing concentrations of IS at 37 °C for up to 9 days. Afterwards, ice-cold RIPA Lysis Buffer was used to lyse the cells at 0, 3, 6 or 9 days for 30 min. The mixture was centrifuged at 14,000× *g*, at 4 °C for 20 min and then quantified the supernatant by BCA Protein Assay Kit. RIPA Lysis Buffer and BCA Protein Assay Kit were purchased from Thermo Scientific (Vienna, Austria).

SDS gels (14–20% acrylamide gradient) were used to separate the proteins. Proteins were separated for 65 min at 120 V (Bio-Rad Laboratories, Hercules, CA, USA), and then transferred to PVDF membranes (Bio-Rad Laboratories, Hercules, CA, USA) using Trans-Blot^®^ Turbo™ Transfer Pack (Bio-Rad Laboratories) and Trans-Blot^®^ Turbo™ Transfer System (Bio-Rad Laboratories) for 7 min at 25 V. A total of 5% skim milk-TBST was used to block the membranes for 2 h, and then the membranes were incubated at 4 °C with the primary antibody overnight and anti-mouse (1:3000, Dako, P0260, Carpinteria, CA, USA) or anti-rabbit (1:3000, Dako, P0448, Carpinteria, CA, USA) secondary antibodies at room temperature for at least 1 h. After that, Clarity Western ECL Blotting Substrate was used to treat the membrane as directed by the manufacturer (Bio-Rad Laboratories, Hercules, CA, USA). Then, the bands were imaged using the ChemiDocTM MP Imaging System. To calculate the density of the bands, Image Lab software (version 6.0.1, Bio-Rad Laboratories, Hercules, CA, USA) was employed. The dilution of primary antibodies was 1:1000. The antibody to detect p21 was purchased from Cell Signaling Technology (Leiden, The Netherlands), and the antibody for LaminB1 was purchased at Abcam (Cambridge, UK).

### 4.5. Real-Time PCR

Total RNA from ciPTEC-OAT1 exposed to IS at different time points was isolated using the RNeasy Mini kit (Qiagen, Venlo, The Netherlands) according to the manufacturer’s instructions. RNA quantity was determined using Nanodrop 2000 (Thermo-Fisher, Waltham, MA, USA). A total of 800 ng mRNA per sample was used to synthesize cDNA by employing the iScriptTM Reverse Transcription Supermix (Bio-Rad Laboratories, Hercules, CA, USA) and the T100™ Thermal Cycler (Bio-Rad Laboratories, Hercules, CA, USA). Real-time PCR was performed using the CFX96TM Real-Time PCR Detection System and the iQ SYBR^®^ Green Supermix (Bio-Rad Laboratories, Hercules, CA, USA) following the manufacturer’s instructions (Bio-Rad Laboratories, Hercules, CA, USA). β-actin was used as a housekeeping gene for normalization. Relative expression levels were calculated using the ΔΔCT method. The primers of different genes are listed as follows (Table 1).

### 4.6. ELISA

Following exposure, cell culture supernatants were collected, centrifuged at 240× *g*, at 4 °C for 10 min and stored at −20 °C until further analysis. The following ELISA Kits from Invitrogen were used according to the manufacturer’s instructions to measure the concentration of SASP factors in the culture supernatants: IL-6 (88-7066-88, Invitrogen, Carlsbad, CA, USA), IL-8 (88-8086-88, Invitrogen, Carlsbad, CA, USA) and IL-1β (88-7261-88, Invitrogen, Carlsbad, CA, USA).

### 4.7. RNA-Seq Differential Gene Expression Analysis

Cells were cultured in the presence or absence of 200 µM IS for 0, 1, 3 and 9 days, and each experiment was performed in triplicate. Cells were directly lysed in 400 µL RLT buffer (Cat. No. 79216, QIAGEN) and stored at −80 °C before RNA extraction on a QIAsymphony isolation robot using the QIAsymphony RNA Kit (931636, QIAGEN) and miRNA CT 400 protocol. RNA quality was checked with the Agilent Fragment Analyzer 5300 system using the RNA Kit (15 nt) (Cat. DNF-471-1000, Agilent, Santa Clara, CA, USA) and RNA quantity was measured with the Invitrogen™ Qubit Flex™ Fluorometer using the Qubit RNA HS Assay Kit (Cat. Q32855). For each sample, 100 ng of total RNA was used to prepare TruSeq Stranded mRNA libraries (Cat. 20020594, Illumina, San Diego, CA, USA) following the manufacturers protocol and with custom 384 xGen UDI-UMI adapters from IDT. Libraries were checked with the Fragment Analyzer system dsDNA 910 Reagent Kit (35–1500 bp) (Cat. DNF-910-K1000, Agilent) and with Qubit dsDNA HS Assay Kit (Cat. Q32854, Invitrogen). Sample libraries were pooled equimolarly before sequencing on a Nextseq2000 (Illumina) using a P2 flowcell with 50 bp paired-end reads, resulting in an average of 20 million reads/sample.

RNA-Seq analysis Quality control on the sequence reads from the raw FASTQ files was done with FastQC (v0.11.8). TrimGalore (v0.6.5) as used to trim reads based on quality and adapter presence, after which FastQC was again used to check the resulting quality. rRNA reads were filtered out using SortMeRNA (v4.3.3), after which the resulting reads were aligned to the reference genome fasta (GCA_000001405.15_GRCh38_no_alt_analysis_set.fna) using the STAR (v2.7.3a) aligner. Follow up QC on the mapped (bam) files was done using Sambamba (v0.7.0), RSeQC (v3.0.1) and PreSeq (v2.0.3). Readcounts were then generated using the Subread FeatureCounts module (v2.0.0) with the Homo_sapiens.GRCh38.106.ncbi.gtf file as an annotation, after which normalization was done using the R-package edgeR (v3.28). Data were analysed using integrated Differential Expression and Pathway analysis (iDEP) [70]. The Kyoto Encyclopedia of Genes and Genomes (KEGG) [71] and MSigDB Hallmark [72] were used to perform the enrichment analysis of the differentially expressed genes (DEGs). The RNAseq raw read counts can be found in Appendix A. Data visualization was prepared by https://www.bioinformatics.com.cn/srplot (last accessed on 22 October 2022), a free online platform for data analysis and visualization.

### 4.8. Statistics

All data analysis and statistics were performed using GraphPad Prism (version 9.3.0; GraphPad software, La Jolla, CA, USA), and expressed as mean ± SEM. Dunnett’s multiple comparison test was employed after the one-way ANOVA to compare different groups. It was considered significant at *p* < 0.05.

## Figures and Tables

**Figure 1 toxins-15-00242-f001:**
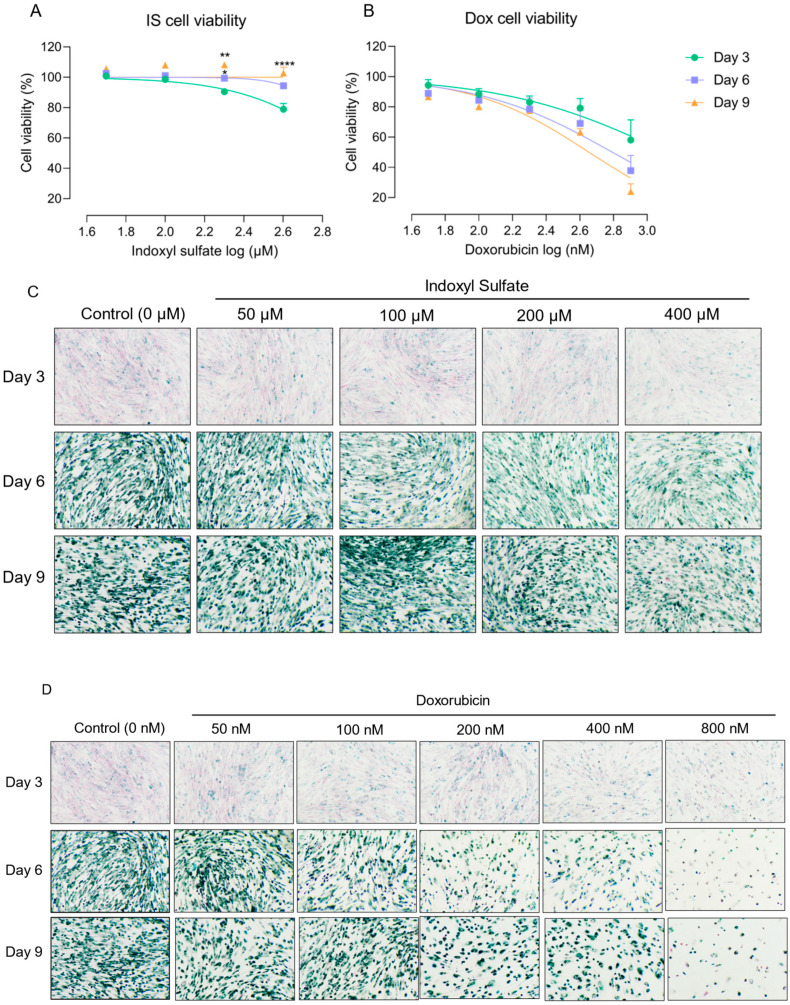
Cellular tolerance against indoxyl sulfate (IS) increased over time, while doxorubicin (Dox) reduced cell viability dose- and time-dependently. Cell viability of ciPTEC-OAT1 exposure to 100 μL medium increased at 0, 3, 6 or 9 days under a non-permissive temperature of 37 °C: (**A**) indoxyl sulfate (IS) and (**B**) doxorubicin (Dox). IC_50_ values for IS are over 400 μM at all time points; IC_50_ values for Dox at Day 3, Day 6 and Day 9 are over 800 nM, 600 nM and 400 nM, respectively. SA-gal staining in ciPTEC-OAT1 cultures grown for 3, 6 and 9 days at 37 °C with various doses: (**C**) IS (50 µM, 100 µM, 200 µM or 400 µM) and (**D**) Dox (50 nM, 100 nM, 200 nM, 400 nM and 800 nM). * *p* < 0.05, ** *p* < 0.01, **** *p* < 0.001 (cell viability at Day 6 and Day 9 compared to Day 3, Two-way ANOVA, Sidak’s multiple comparison).

**Figure 2 toxins-15-00242-f002:**
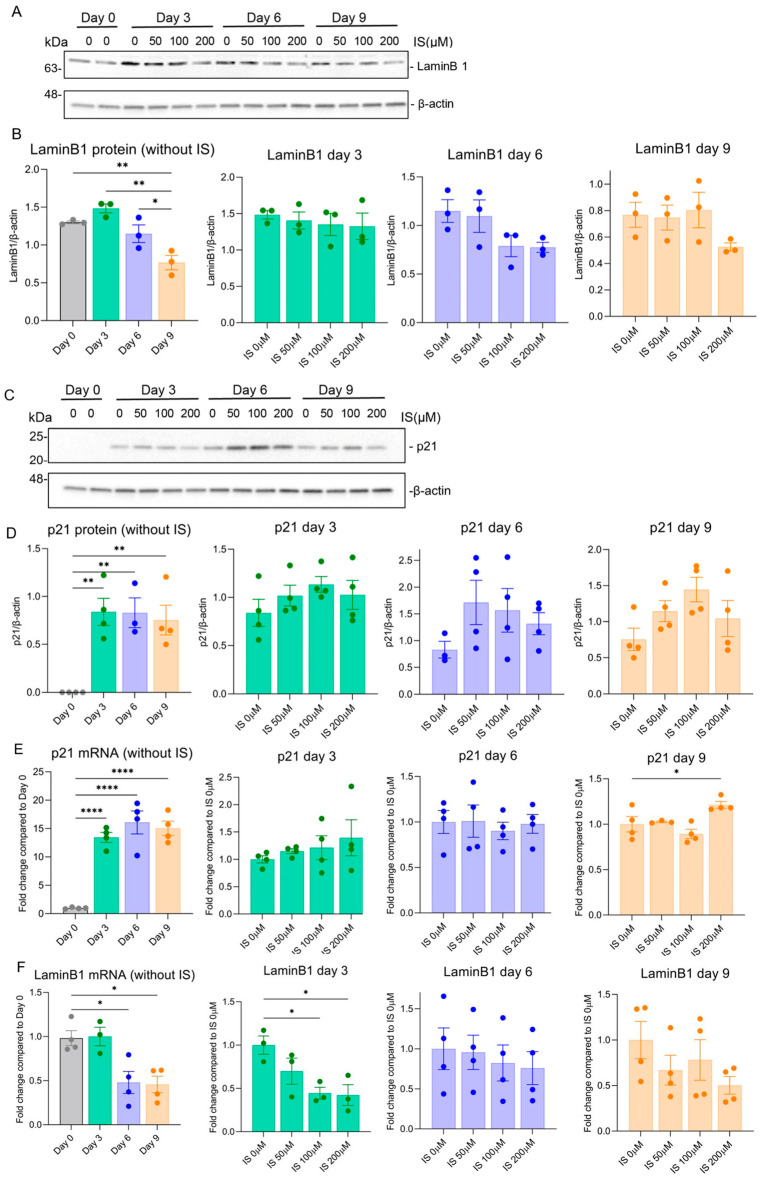
Exposure to IS leads to sustained common senescence markers expression in ciPTEC-OAT1 at a non-permissive temperature of 37 °C. The ciPTEC were exposed to IS (50 µM, 100 µM or 200 µM) for 0, 3, 6 and 9 days. Representative Western blots showing expression of (**A**) LaminB1 and (**C**) p21. Relative protein expression levels of (**B**) LaminB1 and (**D**) p21 over time (Day 0 to Day 9). (**E**) Gene expression levels of p21 and (**F**) LaminB1 over time (Day 0 to Day 9). At least three independent experiments were performed in triplicates. * *p* < 0.05, ** *p* < 0.01, **** *p* < 0.0001 (secreted levels at Days 3, 6, or 9 compared to Day 0 or the non-treatment group at 37 °C; one-way ANOVA, Dunnett’s multiple comparison).

**Figure 3 toxins-15-00242-f003:**
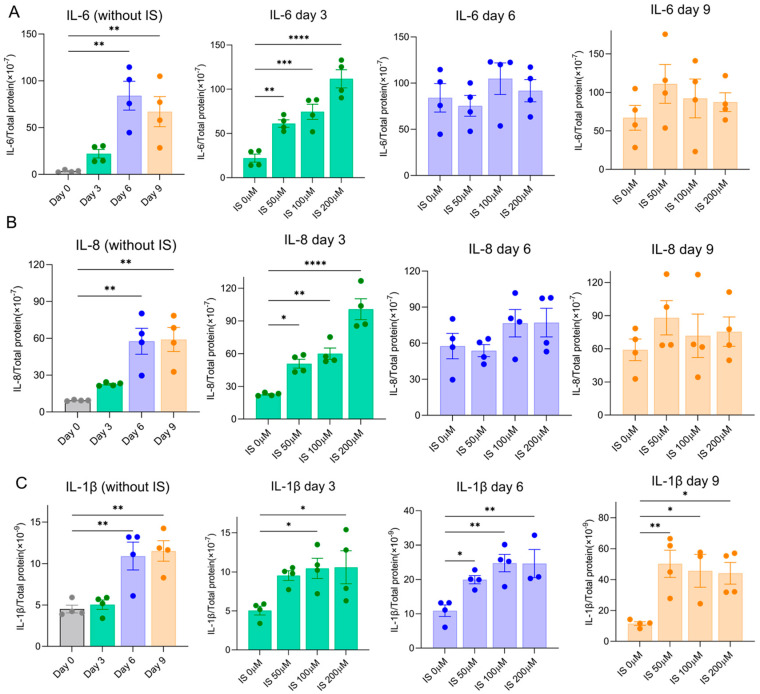
IS accelerated SASP factors secretion by ciPTEC-OAT1 under a non-permissive temperature of 37 °C. Release of (**A**) IL-6, (**B**) IL-8 and (**C**) IL-1β by ciPTEC-OAT1 over time. The concentration is presented as pg/mL and is normalized for total protein (µg/mL). Triplicates of four independent experiments were performed. * *p* < 0.05, ** *p* < 0.01, *** *p* < 0.001, **** *p* < 0.0001 (secreted levels at Days 3, 6 or 9 compared to Day 0 or the non-treatment group at 37 °C; one-way ANOVA, Dunnett’s multiple comparison).

**Figure 4 toxins-15-00242-f004:**
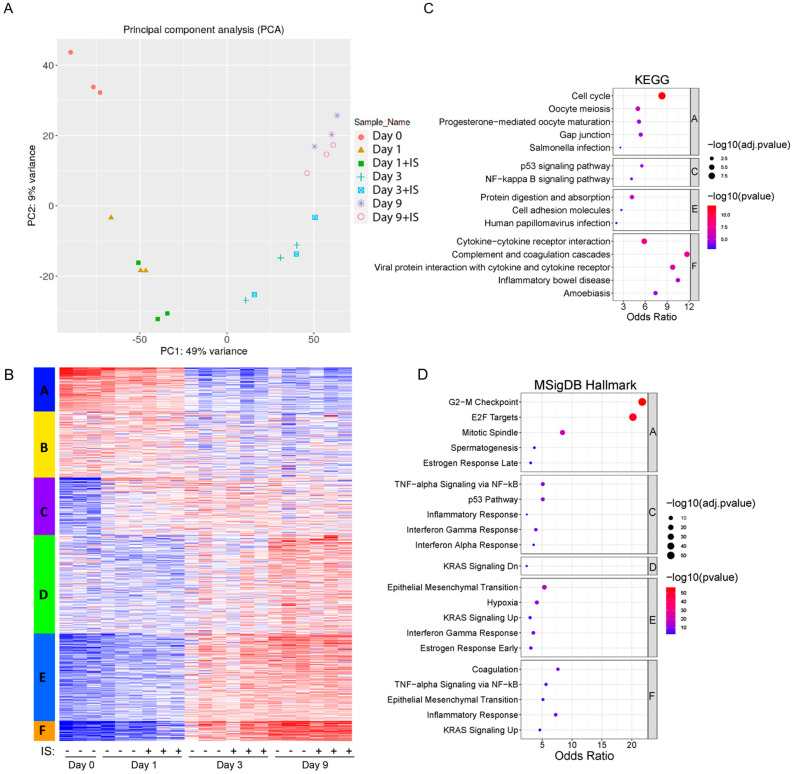
Transcriptomics shows activation of senescence-related pathways over time: (**A**) Principal Component Analysis (PCA) plots from the transcriptome of ciPTEC-OAT1. Different colours and different shapes correspond to different time points and treatments (day 0, and day 1, 3, and 9 in the presence or absence of IS (200 µM). (**B**) K-means clustering of RNA-seq among different samples from 5000 most variable genes. (**C**) Gene networks identified through Kyoto Encyclopedia of Genes and Genomes (KEGG) of top 5 pathways in different clusters, adjusted *p* (adj.p) < 0.05. (**D**) Enrichment analysis of top 5 pathways in different clusters through MSigDB Hallmark, adj.p < 0.05.

**Figure 5 toxins-15-00242-f005:**
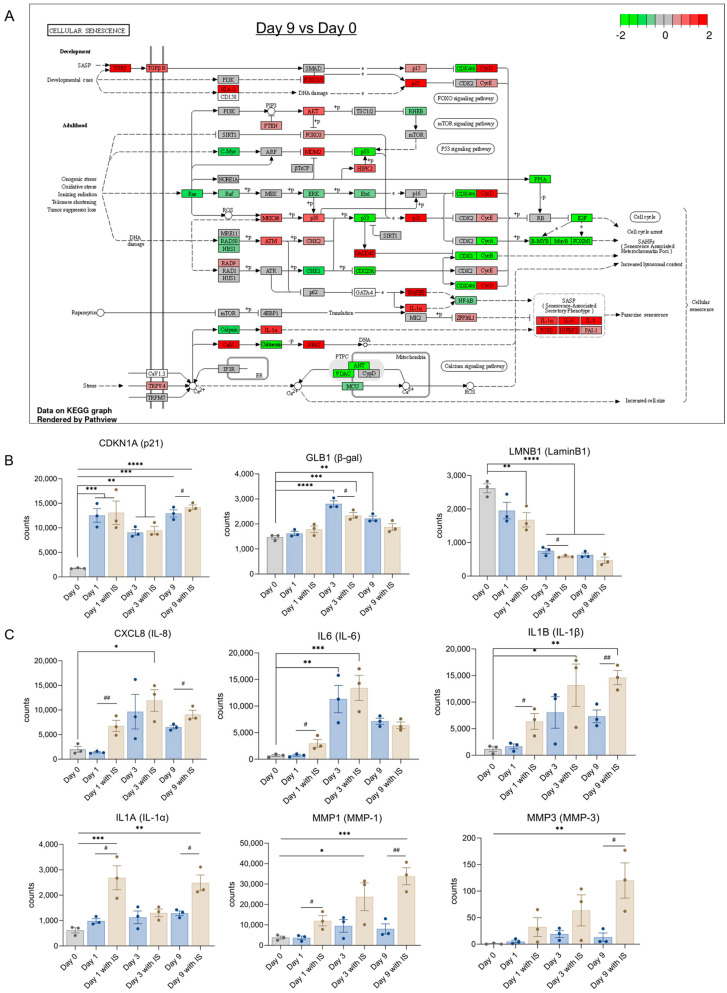
IS accelerates senescence by regulating senescence markers: (**A**) Schematic representation of the cellular senescence pathway at Day 9 compared to Day 0, with indicated upregulation (red) and downregulation (green) of different genes involved in the process. Gene expression levels of (**B**) common senescence markers and (**C**) common SASP factors. * *p* < 0.05, ** *p* < 0.01, *** *p* < 0.001, **** *p* < 0.0001 (secreted levels at Days 3, 6 or 9 compared to Day 0 at 37 °C; one-way ANOVA, Dunnett’s multiple comparison). # *p* < 0.05, ## *p* < 0.01 (secreted levels of the IS-treated group compared to the non-treated group at the same day; Unpaired *t* test).

**Figure 6 toxins-15-00242-f006:**
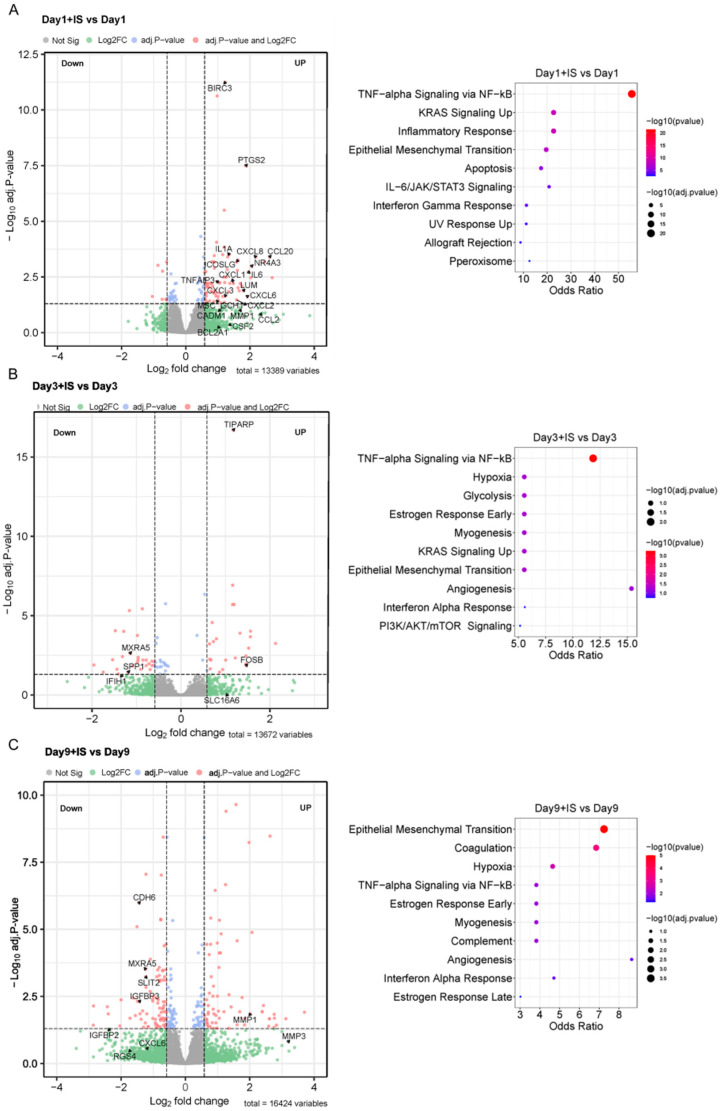
IS induces expression of SASP factors and EMT markers: (**A**–**C**) Volcano plots of differentially expressed genes (DEGs) at different time points, and the top 10 enrichment pathways from MSigDB Hallmark of DEGs in IS-treatment groups compared to non-treatment groups. Volcano plots are representative of DEGs of ciPTEC-OAT1 cultured at 37 °C for 1 to 9 days with or without IS (200 µM) exposure (adj.p < 0.05, |log2 fold change| > 1.5). The most significant DEGs are involved in TNF-alpha Signaling via NF-ĸB and Epithelial Mesenchymal Transition pathway from MSigDB Hallmark.

**Figure 7 toxins-15-00242-f007:**
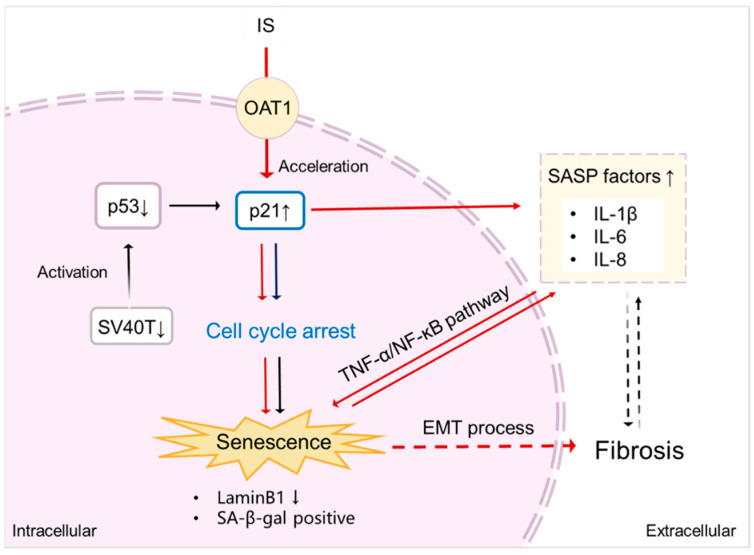
Proposed scheme of senescence acceleration by IS in ciPTEC-OAT1. Downregulation of SV40T and p53 (at 37 °C) leads to transcriptional upregulation of p21, thus inducing cell cycle arrest and eventually causing senescence [34]. After being taken up by the proximal tubule cells via the OAT1 transporter, IS promotes the expression of p21, which results in cell cycle arrest. Moreover, IS specifically upregulates various SASP factors, further accelerating senescence through the TNF-α/NF-ĸB pathway. The senescence process is potentiated by SASP subsequent promotion of fibrosis through the EMT process. Senescent cells show typical SA-β-gal activity and downregulation of LaminB1, characterizing senescence. Red arrows indicate the senescence process accelerated by IS; ↑ upregulation; ↓ downregulation.

**Table 1 toxins-15-00242-t001:** Primers used for real-time polymerase chain reaction.

Gene	Forward Primer	Reverse Primer
p21 (CDKN1A)	TGTCCGTCAGAACCCATGC	AAAGTCGAAGTTCCATCGCTC
laminB1 (LMNB1)	AAGCATGAAACGCGCTTGG	AGTTTGGCATGGTAAGTCTGC
β-actin (ACTB)	CATGTACGTTGCTATCCAGGC	CTCCTTAATGTCACGCACGAT

## Data Availability

RNAseq read counts used for Differential Expression and Pathway analysis have been provided as supplementary data to this manuscript. The RNAseq raw FASTQ files have not been made publicly available for privacy reasons and risk of re-identification of the donor.

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
