# Peer review of "The Uremic Toxin Indoxyl Sulfate Accelerates Senescence in Kidney Proximal Tubule Cells"

_toxins, 2023, doi:10.3390/toxins15040242_

Round 1
Reviewer 1 Report
This paper reports the uremic toxin indoxyl sulfate accelerates senescence in kidney proximal tubule cells. A series of experiments including cell viability assay, SA-β-gal staining assay, Western blot, real-time PCR, ELISA, and transcriptome were conducted to support the title viewpoint. The experimental results look good and reasonable. This finding is not very exciting, because indoxyl sulfate accelerates sencescence in kindney proximal tubule cells is known. But this study may help people to know more about the role of the uremic toxin indoxyl sulfate and the senescence mechanism accelerated by indoxyl sulfate in ciPTEC-OAT1
For example, it was reported that uraemic toxins (including indoxyl sulfate) can induce proximal tubular injury via organic anion transporter 1-mediated uptake (see: Br J Pharmacol, 2002, 135(2): 555, https://doi.org/10.1038/sj.bjp.0704482). This paper investigated the toxic effects of organic anions which characteristically appeared in the patients with progressive renal disease on renal proximal tubular cells expressing human organic anion transporter (hOAT) 1. It also got a conclusion that organic anions present in the serum of patients with progressive renal disease may cause proximal tubular injury via hOAT1-mediated uptake.
Another paper (See: PNAS, 116 (32) 16105-16110,https://doi.org/10.1073/pnas.1821809116) also reported that in the model of human renal proximal tubular cells, indophenol sulfate stimulated the secretion of organic anion (OAT-1) through AhR and EGFR signals.
However, the two closely relevant references were not cited in the current manuscript.
Author Response
Response: We thank the reviewer for appreciating our work and for the recommendation of the relevant references, including our own PNAS paper on the remote sensing and signalling by IS. As the reviewer mentioned, our results are more related the mechanism of IS-induced senescence. We included the references in the revised manuscript to improve the background introduction of the topic (lines 70 to 72).
Reviewer 2 Report
Submission ID toxins-2258599
The aim of the manuscript entitled “The uremic toxin indoxyl sulfate accelerates senescence in kidney proximal tubule cells” was to investigate whether indoxyl sulfate (IS) accelerates senescence in conditionally immortalized proximal tubule epithelial cells overexpressing the organic anion transporter 1 and promotes kidney fibrosis.
The study was conducted on ciPTEC-OAT1 cell model, which was previously shown by the authors to be useful in studies of kidney senescence. Cell viability assay, SA-β-gal staining assay, and gene expression analysis were employed in the study. The manuscript presents interesting findings however should be revised to be published. My suggestions for the revision are the following:
- Regarding the chapter Proximal tubule cells show increased cellular tolerance against indoxyl sulfate over time: One or two sentences of introduction should be included at the beginning of the chapter. To show the increased tolerance of cells to IS over time a graph presenting the cell viability as a function of time, for different concentrations of IS should be presented. Additionally, on Figure 1 A and B, the title of the y axis should be modified by removing 100 (cell viability is presented as %, and not as 100%). In the present form Figure 1A has wrong marks, as point 1.6 is moved to the right and instead on being over 1.6 on the axis is between 1.6 and 1.8. The first sentence of this chapter is unclear and should be reformulated. IC50 was not estimated correctly as at used concentrations of IS the cell viability was not below even 70%. The Authors should use higher concentrations of IS to decrease cell viability below 50%. Please provide rationale for the use of doxorubicin.
- With regard to chapter Indoxyl sulfate exposure results in sustained expression of common senescence markers LaminB1 mRNA levels is shown on 2F instead on 2E figure.
- Page 11, lines 230-233 – rewrite the sentence.
- Taking into account the results shown in Figure 2, IS does not significantly decrease LaminB1 protein level and only decreases significantly LaminB1 mRNA level on day 3, so there is an overstatement in the discussion on Page 11, line 229. Please rewrite.
- As shown in Figure 2, p53 does not significantly increase after IS treatment with an exception of p53 mRNA increased by IS at the highest tested concentration at day 9. Other differences did not reach statistical significance, so we may only speak about a tendency and not significant change.
Author Response
Response: We thank the reviewer for these constructive comments for which you can find our response:
- We included additional sentences to introduce this chapter and modified the title of the y-axis in the revised manuscript (lines 96 to 98). We agree with the reviewer that estimating a more accurate IC50 for IS, higher concentrations should be used. Indeed, the cell viability was not below 70%, which confirms our previous findings that indoxyl sulfate has an IC50 of approx. 2.0 mM in presence of inhibitors of efflux (Jansen et al. Fig. 2)[1]. The current study focused on investigating IS concentrations that are clinically relevant (up to around 200 μM [2]) and not on direct, acute toxicity. The lowest concentration of IS is 200 μM. We transformed the concentration into log(μM) in figure 1A, so the first point of IS (around 1.7) refers to log(200 μM). For clarity, we reformatted the figures and offset the X and Y axes. Doxorubicin was reported as a senescence inducer in previous research[3,4], so it was employed as a positive control to induce senescence. However, the results suggested that doxorubicin not only induced senescence but also eliminated senescent cells. Meanwhile, IS induced and accumulated senescent cells. We now include an explanation and the reference in our revised manuscript (lines 98 to 101, and lines 108 to 109).
- We thank the reviewer for the observation, now we corrected this point in the revised manuscript. (line 124).
3, 4. Sentences are rewritten. (lines 248 to 252).
- We have tuned down our statements in the revised version of manuscript. (lines 124 to 125).
Reviewer 3 Report
Authors investigated IS as a uremic toxin to contribute development of fibrosis through senescent cells. In the present manuport, authors generally well approached to reach their conclusion.
However, doxorubicin-induced renal injuries are not a universal model, but a specific model related to drug-associated interstitial injuries in kidney. Therefore, it would be better to decrease their messages or might be required to add the associated data using remnant kidney animal model which is recognized as chronic and progressive kidney disease model.
The other major comment
1. In the Western blotting photo of Figure 2A, lanes of 0 of IS showed more high densities in βactine as a reference, didn’t they? I could not find that the present results of Lamin B1 might support their results. It would be better to replace another Western blotting results since authors tried three times at least from their description in text, Or I recommend to perform this experiment again and to change more resorbable results.
Author Response
Response: We thank the reviewer for the constructive comments. As the reviewer mentioned, doxorubicin-induced injury is a specific model related to drug-associated interstitial injuries in kidney. But as a chemotherapeutic agent, doxorubicin was also reported as an inducer of senescence [3,4]. When we started, the drug was employed as a positive control to induce senescence. However, it turned out to be a negative control as the results suggested that doxorubicin not only induces senescence but also eliminates senescent cells, while our data further imply that IS-induced senescence is more stable. To clarify this, we added an explanation in the revised manuscript (lines 100 to 101, and lines 108 to 109).
We thank the reviewer for valid remarks. We replaced the image with other Western blotting results of LaminB1 in the revised manuscript. Additionally, all Western blot images are included in Supplementary material (WB original).